# Considerations for ensuring safety during telerehabilitation of people with stroke. A protocol for a scoping review

**Ruvistay Gutierrez-Arias** [1,2] *, **Camila González-Mondaca** [2☯], **Vinka Marinkovic-Riffo** [2☯], **Marietta Ortiz-Puebla** [2☯], **Fernanda Paillán-Reyes** [2☯], **Pamela Seron** [3,4]

1 Servicio de Medicina Física y Rehabilitación, Unidad de Kinesiología, Instituto Nacional del Tórax, Santiago, Chile, 2 Exercise and Rehabilitation Sciences Institute, School of Physical Therapy, Faculty of Rehabilitation Sciences, Universidad Andres Bello, Santiago, Chile, 3 Centro de Excelencia CIGES, Universidad de La Frontera, Temuco, Chile, 4 Departamento de Ciencias de la Rehabilitación, Facultad de Medicina, Universidad de La Frontera, Temuco, Chile

☯ These authors contributed equally to this work.
* ruvistay.gutierrez@gmail.com

**Data Availability Statement:** No datasets were generated or analysed during the current study. All

## Abstract

### Introduction

Exercise interventions have a positive impact on people with stroke. However, access to exercise interventions is variable, and there may be a delay in the start of rehabilitation. Telerehabilitation has enabled the delivery of exercise interventions replacing the traditional face-to-face approach. Aspects related to the safety of people with stroke should be considered to avoid adverse events during the delivery of exercise interventions remotely. However, such information is scattered in the literature, and the detail with which measures taken during the implementation of exercise interventions for people with stroke are reported is unknown.

### Objective

To summarise measures or aspects targeted at reducing the incidence of adverse events during the delivery of exercise interventions through telerehabilitation in patients after stroke.

### Materials and methods

A scoping review will be conducted. A systematic search in MEDLINE-Ovid, Embase-Ovid CENTRAL, CINAHL Complete (EBSCOhost), and other resources will be carried out. We will include primary studies, published in full text in any language, involving people with stroke who undergo telerehabilitation where exercise is the main component. Two reviewers will independently select studies and extract data, and disagreements will be resolved by consensus or a third reviewer. The results will be reported in a narrative form, using tables and figures to support them.

relevant data from this study will be made available upon study completion.

**Funding:** The authors received no specific funding for this work.

**Competing interests:** The authors have declared that no competing interests exist.

## Discussion

To implement this strategy within rehabilitation services, one of the first aspects to be solved is to ensure the safety of people. The results of this scoping review could contribute an information base for clinicians and decision-makers when designing remotely delivered exercise intervention programs.

## Registration number

INPLASY202290104.

## Introduction

Stroke is a disease characterized by a focal deficit due to acute injury to the central nervous system [1]. This injury of vascular origin includes cerebral infarction and intracerebral or subarachnoid hemorrhage [1]. Stroke is the second leading cause of death and disability worldwide, resulting in a high burden of disease, especially in low- and middle-income countries [2]. The reported prevalence of stroke globally in 2016 was 80.1 million (95% CI 74.1–86.3) [3].

The sequelae in people with stroke are diverse [4]. Regarding physical function post-stroke, functional impairment of the upper and lower extremities is common, which may be due to weakness or paralysis, sensory loss, spasticity, and abnormal motor synergies [5]. In addition, a near 15% prevalence of sarcopenia has been found in people with stroke [6]. Gait impairment has been observed in a high percentage of people with stroke [7, 8], a dysfunction that may persist despite rehabilitation [9].

More than 50% of people with stroke may experience limitations in activities such as shopping, housework, and difficulty reintegrating into community life within 6 months [10]. These restrictions can result in a diminished health-related quality of life [10–12].

Rehabilitation is a multiple and comprehensive intervention, of which exercise is one of the main components. Exercise interventions have a positive impact on people with stroke [13], with a small to moderate effect on the quality of life (standardized mean difference of -0.23 (95% CI, -0.40 to -0.07)) [14], and an increase in cognitive performance [15]. However, access to exercise interventions is variable [16–18], and there may be a delay in the start of physical rehabilitation [19]. This could be made worse in contexts such as the COVID-19 pandemic, however, strategies such as telerehabilitation were implemented, which contributed to continued service delivery [20].

Telerehabilitation has enabled the delivery of exercise interventions replacing the traditional face-to-face approach in patient-rehabilitator interaction [21]. The potential for telerehabilitation to achieve similar clinical outcomes to traditional rehabilitation, and better than no rehabilitation at all [22–25], should prompt healthcare facilities to evaluate implementing remote delivery of exercise interventions for people with stroke. Telerehabilitation in stroke patients can be used to deliver interventions aimed at improving cognitive [26], physical [22, 27], speech [28] and swallowing function [29].

For this, in addition to logistics and costs, aspects related to the safety of people with stroke should be considered to avoid adverse events during the delivery of exercise interventions. This information could be reported from studies that have evaluated the feasibility, safety, or effectiveness of telerehabilitation in this population. However, such information is scattered in

the literature, and the detail with which measures taken during the implementation of exercise interventions for people with stroke are reported is unknown. Therefore, this study aims to summarise measures or aspects targeted at reducing the incidence of adverse events during the delivery of exercise interventions through telerehabilitation in patients after stroke.

## Materials and methods

A scoping review will be conducted following the updated recommendations of the Joanna Briggs Institute (JBI) [30]. The protocol for this review was registered on the International Platform of Registered Systematic Review and Meta-analysis Protocols (INPLASY) under the number INPLASY202290104, and it was reported following Preferred Reporting Items for Systematic Review and Meta-analysis Protocols (PRISMA-P) (S1 Checklist) [31]. The results will be reported following the Extension for Scoping Reviews of the Preferred Reporting Items for Systematic Reviews and Meta-analyses statement (PRISMA-ScR) [32].

### Search strategy

A systematic search of MEDLINE, through the Ovid platform; Embase, through the Ovid platform; Cochrane Collaboration Central Register of Controlled Trials (CENTRAL), through the Cochrane Library; and Cumulative Index of Nursing and Allied Literature Complete (CINAHL Complete), through the EBSCOhost platform. The strategy will consider a sensitive approach and the use of controlled language (MeSH, EMTREE, CINAHL Subject Heading) and natural language. The strategy to be used for MEDLINE-Ovid will be adapted to construct the search in the other databases (Table 1).

The search will not be limited by publication date, publication status, or the language of the studies. Filters will be applied to the different strategies to exclude systematic reviews, with and without meta-analysis, from the search results.

In addition, studies included in systematic reviews aimed at evaluating the effectiveness of telerehabilitation in stroke patients will be screened.

### Eligibility criteria

Eligibility criteria for study selection will be divided into participants or populations included in the studies, the concept or phenomenon involved, and the context in which the studies were conducted (i.e. population, concept and context (PCC) framework) [30]. In addition, the design of the studies will be considered for inclusion in this review.

**Participants.**   This review will include studies involving people with stroke, regardless of type, cause, the time course of the disease and sequelae caused.

**Concept.**   This review will include studies where exercise interventions are delivered through telerehabilitation. Exercise interventions shall be understood as a subcategory of physical activity that is planned, structured, repetitive, and purposefully focused on improving or maintaining one or more components of physical fitness [33].

Interventions may be delivered synchronously, asynchronously, or mixed. Studies in which the professional delivering the intervention has face-to-face contact with the person with stroke to conduct assessments or educational sessions on the exercises to be performed (hybrid programmes) will not be excluded. The methodology for delivering interventions remotely may be by videoconferencing, mobile phone, pre-recorded videos, text message reminders, web platforms or apps.

**Context.**   This review will include studies in which people with stroke perform the exercises as prescribed outside the hospital setting, either at home or in community centres, and

**Table 1. Search strategy for MEDLINE using the Ovid platform.**

| N° | Search term |
|---|---|
| 1 | exp Cerebrovascular Disorders/ |
| 2 | exp basal ganglia cerebrovascular disease/ |
| 3 | exp brain ischemia/ |
| 4 | exp carotid artery diseases/ |
| 5 | exp cerebral small vessel diseases/ |
| 6 | exp intracranial arterial diseases/ |
| 7 | exp "intracranial embolism and thrombosis"/ |
| 8 | exp intracranial hemorrhages/ |
| 9 | exp stroke/ |
| 10 | exp brain infarction/ |
| 11 | exp stroke, lacunar/ |
| 12 | exp vasospasm, intracranial/ |
| 13 | exp vertebral artery dissection/ |
| 14 | (stroke$ or poststroke or apoplex$ or cerebral vasc$ or brain vasc$ or cerebrovasc$ or cva$ or SAH).ti,ab. |
| 15 | ((brain$ or cerebr$ or cerebell$ or vertebrobasil$ or hemispher$ or intracran$ or intracerebral or infratentorial or supratentorial or middle cerebral artery or MCA$ or anterior circulation or posterior circulation or basilar artery or vertebral artery or space-occupying) adj5 (isch?emi$ or infarct$ or thrombo$ or emboli$ or occlus$ or hypoxi$)).ti,ab. |
| 16 | ((brain$ or cerebr$ or cerebell$ or intracerebral or intracran$ or parenchymal or intraparenchymal or intraventricular or infratentorial or supratentorial or basal gangli$ or putaminal or putamen or posterior fossa or hemispher$ or subarachnoid) adj5 (h?emorrhag$ or h?ematoma$ or bleed$)).tw. |
| 17 | or/1-16 |
| 18 | Telerehabilitation/ |
| 19 | exp Videoconferencing/ |
| 20 | telecommunications/ |
| 21 | Remote Consultation/ |
| 22 | (telemetry or telerehab$ or tele-rehab$ or telehealth or tele-health or telehomecare or telehomecare or telecoaching or tele-coaching or videoconference$ or video-conferenc$ or videoconsultation or video-consultation or teleconference$ or tele-conference$ or teleconsultation or tele-consultation or telecare or telecare).ti,ab. |
| 23 | (ehealth or e-health or "mobile health" or mhealth or m-health).ti,ab. |
| 24 | ((remote$ or distance$ or distant) adj5 (rehab$ or therap$ or treatment or consultation)).ti,ab. |
| 25 | ((rehab$ or therap$ or treatment or consultation) adj5 (telephone$ or phone$ or video$ or internet$ or computer$ or modem or web$ or email)).ti,ab. |
| 26 | or/18-25 |
| 27 | 17 and 26 |
| 28 | systematic review/ |
| 29 | meta-analysis/ |
| 30 | (meta analy$ or metanaly$ or metaanaly$).ti,ab. |
| 31 | ((systematic or evidence) adj2 (review$ or overview$)).ti,ab. |
| 32 | (reference list$ or bibliograph$ or hand search$ or manual search$ or relevant journals).ab. |
| 33 | (medline or pubmed or cochrane or embase or psychlit or psyclit or psychinfo or psycinfo or cinahl or science citation index or bids or cancerlit).ab. |
| 34 | cochrane.jw. |
| 35 | or/28-34 |
| 36 | 27 not 35 |
| 37 | animals/ not humans/ |
| 38 | 36 not 37 |

can be performed individually or in groups. The therapist may or may not be remotely supervising the exercise sessions.

**Study designs.** Primary studies (randomised or non-randomised clinical trials, cohort studies, case-control, cross-sectional, and case reports, among others) will be included. In terms of publication status, studies in which only the abstract is available, such as those presented in conference proceedings, will be excluded. In addition, survey-based studies assessing barriers and facilitators of telerehabilitation in people with stroke will be excluded.

Studies that do not report a clinical outcome, such as level of function, quality of life, muscle strength, safety, among others, as well as the level of satisfaction, will be excluded.

The language as well as the publication date of the studies will not limit their inclusion.

**Selection of studies.** Once the search for studies has been conducted, titles and abstracts will be independently screened by two research team members, who will discard studies irrelevant to this review. Subsequently, the full texts of the potential studies to be included will be analyzed to determine which articles meet all the eligibility criteria. The Rayyan® app will be used for this stage [34]. This tool improves the efficiency of the study selection process both in the title-abstract and full-text screening phase, eliminates duplicate records, facilitates the construction of the study selection diagram and helps to resolve disagreements.

**Information extraction.** Two reviewers will independently extract information from the included studies. An extraction form specifically designed to meet the objectives of this review will be used and will be developed in a Microsoft Excel® spreadsheet.

The information to be extracted will include aspects related to the characteristics of the publications and studies, as well as the population, interventions delivered (type of exercise and technological media used), and outcomes assessed (Table 2). In studies reporting on measures implemented to prevent adverse events during telerehabilitation sessions, detailed information will be extracted, such as the time of delivery of the intervention, the professional involved, the specific measure implemented, among others.

In case the information presented by the studies is unclear or missing, the authors of the studies will not be contacted, as this scoping review also aims to assess the completeness of the reporting of the studies.

Reviewers involved in study selection and data extraction should have clinical experience in the care of stroke patients through tele-rehabilitation, as well as training in conducting evidence synthesis studies. A novice reviewer will always be paired with a senior reviewer.

For both the selection and extraction phases, disagreements will be resolved by consensus. If this is not achieved a third reviewer will make the final decision.

**Table 2. Information to be extracted from the included studies.**

| Information | Description |
|---|---|
| Identification of the studies | Title of the study, name of the journal, year of publication, authors' names, and authors' nationality. |
| Classification of studies | Aim of the study, study design, inclusion and exclusion criteria, and level of evidence according to the Oxford scale [35]. |
| Exercise interventions | Type of exercise, dosage, duration of the session and the complete program, and prescribing professional. |
| Method used for telerehabilitation | Synchronicity and technology used to deliver the intervention. |
| Outcomes | List of outcomes assessed and reported in the study, other than adverse events. |
| Adverse events | Incidence and description of adverse events arising from telerehabilitation. |
| Conclusions | Conclusions related to the safety of delivering exercise interventions remotely. |

**Synthesis of information.** The results of the search and selection of studies will be reported through a PRISMA flow chart [36]. In addition, the reasons for the exclusion of full-text evaluated studies will be reported in a table.

The results will be reported in narrative form, and tables and figures will be used to synthesize the information. Waffle charts [37], or similar, will be used to represent the different primary study designs included, and the frequency of reporting of measures implemented to prevent adverse events during telerehabilitation. In addition, the frequency of the different measures implemented will be reported in one or more figures.

## Discussion

Different situations, such as the pandemic caused by COVID-19, may require the restructuring of health services to maintain people's health care, such as rehabilitation interventions [20].

Telerehabilitation appears as an alternative that can contribute to this challenge, especially for people with stroke, due to their high disease burden [2]. To implement this strategy within rehabilitation services, one of the first aspects to be solved is to ensure the safety of people [38, 39].

The results of this scoping review could contribute an information base for clinicians and decision-makers when designing remotely delivered exercise intervention programs. In addition, knowledge gaps will be observed, which will serve as a basis for future research on this important topic.

## Supporting information

**S1 Checklist. PRISMA-P 2015 checklist.**
(DOCX)

## Author Contributions

**Conceptualization:** Ruvistay Gutierrez-Arias, Camila González-Mondaca, Vinka Marinkovic-Riffo, Marietta Ortiz-Puebla, Fernanda Paillán-Reyes.

**Methodology:** Ruvistay Gutierrez-Arias, Pamela Seron.

**Project administration:** Ruvistay Gutierrez-Arias.

**Supervision:** Ruvistay Gutierrez-Arias.

**Writing – original draft:** Ruvistay Gutierrez-Arias.

**Writing – review & editing:** Camila González-Mondaca, Vinka Marinkovic-Riffo, Marietta Ortiz-Puebla, Fernanda Paillán-Reyes, Pamela Seron.

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
