## [Decision Letter · Decision Letter 0]

7 Dec 2022

PONE-D-22-27366Considerations for ensuring safety during telerehabilitation of people with stroke. A protocol for a scoping reviewPLOS ONE

Dear Dr. Gutierrez-Arias,

Thank you for submitting your manuscript to PLOS ONE. After careful consideration, we feel that it has merit but does not fully meet PLOS ONE’s publication criteria as it currently stands. Therefore, we invite you to submit a revised version of the manuscript that addresses the points raised during the review process.

We look forward to receiving your revised manuscript.

Kind regards,

Fatih Özden, PhD

Academic Editor

PLOS ONE

Journal Requirements:

"No funding was received for the development of this scoping review protocol."

Reviewers' comments:

Reviewer's Responses to Questions

**Comments to the Author**

1. Does the manuscript provide a valid rationale for the proposed study, with clearly identified and justified research questions?

Reviewer #1: Yes

Reviewer #2: Yes

2. Is the protocol technically sound and planned in a manner that will lead to a meaningful outcome and allow testing the stated hypotheses?

Reviewer #1: Yes

Reviewer #2: Yes

3. Is the methodology feasible and described in sufficient detail to allow the work to be replicable?

Reviewer #1: Yes

Reviewer #2: Yes

4. Have the authors described where all data underlying the findings will be made available when the study is complete?

Reviewer #1: Yes

Reviewer #2: Yes

5. Is the manuscript presented in an intelligible fashion and written in standard English?

Reviewer #1: Yes

Reviewer #2: Yes

6. Review Comments to the Author

You may also provide optional suggestions and comments to authors that they might find helpful in planning their study.

Reviewer #1: Dear authors,

thank you for the possibility to review your scoping review protocol.

I have some suggestions:

Line: 68: Why only a small or moderate impact on quality of life?

Some sentences are confusing in the wording, e.g. line 70-72. I think also strategies for telerehabilitation had to be adapted? Or does it mean a the implementation of a new concept like telerehabilitation is a positive effect of the pandemic? Then I would describe it in a more positive manner.

Why do you not refer to other diseases with positive impacts due to telerehabilitation to support your research interests? (e.g. Piotrowicz E. et al. doi: 10.1001/jamacardio.2019.5006.

Also I miss, the different kinds of telerehabilitatio, e.g. telerehabilitation in mobility, or speech training

Please write out the abbreviation PCC (line 112)

Line 130: Will they distinguish between the different evidence levels of the included studies?

Line 140: Please describe necassary qualification of reviewers

Line 142: Please describe the benefit of using the Rayyan® app

Line 158: Do not repeat the sentence exactly of line 144-45. Maybe you can write this as overall method in case of disagreements.

Is it planned to contact study investigator in case of missing or confirming information

Which tiime frame of the analysis will be covered?

Please include a bias or evidence grade information of the study in table 2 depending on the kind of study and also of number of included patients in the study.

Reviewer #2: The manuscript is well designed. The authors stated that "The search was not limited by publication date, publication status, or the language of the". Future tense should be preferred for the whole manuscript. Also, authors should give more details on Rayyan software. King Regards

7. PLOS authors have the option to publish the peer review history of their article (what does this mean?). If published, this will include your full peer review and any attached files.

Reviewer #1: **Yes: **Dr. rer. medic. Sandra Prescher

Reviewer #2: No

---

## [Author Response · Author response to Decision Letter 0]

11 Dec 2022

Response to Reviewers

Comment 

Line: 68: Why only a small or moderate impact on quality of life? 

Response

The effect is small to moderate due to the values of the 95% CI limits of the standardized mean difference estimated by the study of Ali et al. Specific numerical results were added in lines 69 and 70 of the revised manuscript.

Comment 

Some sentences are confusing in the wording, e.g. line 70-72. I think also strategies for telerehabilitation had to be adapted? Or does it mean a the implementation of a new concept like telerehabilitation is a positive effect of the pandemic? Then I would describe it in a more positive manner. 

Response

The idea was rephrased to give it a more "positive" look.

In lines 73 to 74 the following sentence was added: “…however, strategies such as telerehabilitation were implemented, which contributed to continued service delivery.”

Comment 

Why do you not refer to other diseases with positive impacts due to telerehabilitation to support your research interests? (e.g. Piotrowicz E. et al. doi: 10.1001/jamacardio.2019.5006. 

Response

The reference suggested by the reviewer and others have been added.

Comment 

Also I miss, the different kinds of telerehabilitatio, e.g. telerehabilitation in mobility, or speech training 

Response

Four functions were added where telerehabilitation could have an impact on people with stroke. 

The following sentence was added to lines 81 to 82: “Telerehabilitation in stroke patients can be used to deliver interventions aimed at improving cognitive [26], physical [22,27], speech [28] and swallowing function [29].”

Comment

Please write out the abbreviation PCC (line 112) 

Response

Abbreviation added on line 117 of the revised manuscript.

Comment

Line 130: Will they distinguish between the different evidence levels of the included studies? 

Response

The level of evidence is reported according to the Oxford scale. This information was added to table 2.

Comment

Line 140: Please describe necassary qualification of reviewers 

Response

The following idea was added to the revised manuscript: “Reviewers involved in study selection and data extraction should have clinical experience in the care of stroke patients through tele-rehabilitation, as well as training in conducting evidence synthesis studies. A novice reviewer will always be paired with a senior reviewer.”

Comment

Line 142: Please describe the benefit of using the Rayyan® app 

Response

The following idea was added to the revised manuscript: “This tool improves the efficiency of the study selection process both in the title-abstract and full-text screening phase, eliminates duplicate records, facilitates the construction of the study selection diagram and helps to resolve disagreements.”

Comment

Line 158: Do not repeat the sentence exactly of line 144-45. Maybe you can write this as overall method in case of disagreements. 

Response

Both sentences were deleted, and one was added. The following idea was added to the revised manuscript: “For both the selection and extraction phases, disagreements will be resolved by consensus. If this is not achieved a third reviewer will make the final decision.”

Comment

Is it planned to contact study investigator in case of missing or confirming information 

Response

No information will be required from the authors of the studies in the cases indicated, as this review also seeks to determine the quality of the reporting of the studies. The following idea was added to the revised manuscript: “In case the information presented by the studies is unclear or missing, the authors of the studies will not be contacted, as this scoping review also aims to assess the completeness of the reporting of the studies.”

Comment

Which tiime frame of the analysis will be covered? 

Response

In the "Study design" sub-section of the eligibility criteria, it is reported that studies will not be restricted for inclusion based on date of publication.

Comment

Please include a bias or evidence grade information of the study in table 2 depending on the kind of study and also of number of included patients in the study. 

Response

The risk of bias and certainty of evidence is assessed when we want to determine how much we can believe the results of studies. In addition, the certainty of evidence is determined for each outcome reported by the study, and not for single studies.

Taking into account the above two points, associated with the fact that scoping reviews do not seek to determine the effectiveness of interventions (that is the task of systematic reviews and meta-analyses) and that the objective of our review is to determine the measures implemented to prevent adverse events during tele-rehabilitation of people with stroke, the assessment of risk of bias and certainty of evidence does not apply to our study.

Comment

The authors stated that "The search was not limited by publication date, publication status, or the language of the". Future tense should be preferred for the whole manuscript. 

Response

The sentence was changed to the future tense.

Comment

Also, authors should give more details on Rayyan software

Response 

The following idea was added to the revised manuscript: “This tool improves the efficiency of the study selection process both in the title-abstract and full-text screening phase, eliminates duplicate records, facilitates the construction of the study selection diagram and helps to resolve disagreements.”

---

## [Decision Letter · Decision Letter 1]

26 Dec 2022

Considerations for ensuring safety during telerehabilitation of people with stroke. A protocol for a scoping review

PONE-D-22-27366R1

Dear Dr. Gutierrez-Arias,

We’re pleased to inform you that your manuscript has been judged scientifically suitable for publication and will be formally accepted for publication once it meets all outstanding technical requirements.

Kind regards,

Fatih Özden, PhD

Academic Editor

PLOS ONE

Additional Editor Comments (optional):

Reviewers' comments:

Reviewer's Responses to Questions

**Comments to the Author**

1. Does the manuscript provide a valid rationale for the proposed study, with clearly identified and justified research questions?

Reviewer #1: Yes

Reviewer #2: Yes

2. Is the protocol technically sound and planned in a manner that will lead to a meaningful outcome and allow testing the stated hypotheses?

Reviewer #1: Yes

Reviewer #2: Yes

3. Is the methodology feasible and described in sufficient detail to allow the work to be replicable?

Reviewer #1: Yes

Reviewer #2: Yes

4. Have the authors described where all data underlying the findings will be made available when the study is complete?

Reviewer #1: Yes

Reviewer #2: Yes

5. Is the manuscript presented in an intelligible fashion and written in standard English?

Reviewer #1: Yes

Reviewer #2: Yes

6. Review Comments to the Author

You may also provide optional suggestions and comments to authors that they might find helpful in planning their study.

Reviewer #1: Thank you for taking my suggestions into consideration. Please please delete the " Translated with www.DeepL.com/Translator (free version)" in line 181. I am not sure if using a direct translation by deepl has to be referenced.

Reviewer #2: I hare reviewed the responses and revisions of the authors. The authors was correctly conducted my minor suggestions

7. PLOS authors have the option to publish the peer review history of their article (what does this mean?). If published, this will include your full peer review and any attached files.

Reviewer #1: **Yes: **Sandra Prescher

Reviewer #2: No

---

## [Editor Report · Acceptance letter]

28 Dec 2022

PONE-D-22-27366R1 

Considerations for ensuring safety during telerehabilitation of people with stroke. A protocol for a scoping review 

Dear Dr. Gutierrez-Arias:

I'm pleased to inform you that your manuscript has been deemed suitable for publication in PLOS ONE. Congratulations! Your manuscript is now with our production department. 

Kind regards, 

on behalf of

Dr. Fatih Özden 

Academic Editor

PLOS ONE